# Neurotransmitters: Potential Targets in Glioblastoma

**DOI:** 10.3390/cancers14163970

**Published:** 2022-08-17

**Authors:** Qiqi Huang, Lishi Chen, Jianhao Liang, Qiongzhen Huang, Haitao Sun

**Affiliations:** 1Clinical Biobank Center, Microbiome Medicine Center, Department of Laboratory Medicine, Zhujiang Hospital and the Second Clinical Medical College, Southern Medical University, Guangzhou 510280, China; 3180010053@i.smu.edu.cn (Q.H.); 3190010070@i.smu.edu.cn (L.C.); 2Neurosurgery Center, The National Key Clinical Specialty, The Engineering Technology Research Center of Education Ministry of China on Diagnosis and Treatment of Cerebrovascular Disease, Guangdong Provincial Key Laboratory on Brain Function Repair and Regeneration, The Neurosurgery Institute of Guangdong Province, Zhujiang Hospital, Southern Medical University, Guangzhou 510280, China; allen15622174002@i.smu.edu.cn; 3Department of Neurosurgery, Medical Center, University of Freiburg, 79106 Freiburg im Breisgau, Germany; qiongzhen.huang@uniklinik-freiburg.de; 4Key Laboratory of Mental Health of the Ministry of Education, Guangdong–Hong Kong–Macao Greater Bay Area Center for Brain Science and Brain–Inspired Intelligence, Southern Medical University, Guangzhou 510515, China

**Keywords:** neurotransmitters, glioblastomas multiforme (GBM), neural microenvironment, GBM microenvironment, targeted therapies, antagonist/agonist

## Abstract

**Simple Summary:**

Aiming to discover potential treatments for GBM, this review connects emerging research on the roles of neurotransmitters in the normal neural and the GBM microenvironments and sheds light on the prospects of their application in the neuropharmacology of GBM. Conventional therapy is blamed for its poor effect, especially in inhibiting tumor recurrence and invasion. Facing this dilemma, we focus on neurotransmitters that modulate GBM initiation, progression and invasion, hoping to provide novel therapy targeting GBM. By analyzing research concerning GBM therapy systematically and scientifically, we discover increasing insights into the regulatory effects of neurotransmitters, some of which have already shown great potential in research in vivo or in vitro. After that, we further summarize the potential drugs in correlation with previously published research. In summary, it is worth expecting that targeting neurotransmitters could be a promising novel pharmacological approach for GBM treatment.

**Abstract:**

For decades, glioblastoma multiforme (GBM), a type of the most lethal brain tumor, has remained a formidable challenge in terms of its treatment. Recently, many novel discoveries have underlined the regulatory roles of neurotransmitters in the microenvironment both physiologically and pathologically. By targeting the receptors synaptically or non-synaptically, neurotransmitters activate multiple signaling pathways. Significantly, many ligands acting on neurotransmitter receptors have shown great potential for inhibiting GBM growth and development, requiring further research. Here, we provide an overview of the most novel advances concerning the role of neurotransmitters in the normal neural and the GBM microenvironments, and discuss potential targeted drugs used for GBM treatment.

## 1. Introduction

Glioblastoma (GBM; Grade IV astrocytoma) is among the most aggressive tumors of the central nervous system (CNS) [1], characterized by fast development and an average survival of only 14 months in patients diagnosed with GBM. Currently, treatment strategies for GBM patients are limited. Radiotherapy and chemotherapy with temozolomide (TMZ) following debulking surgery are the standard treatment options [2]. Although the combination of TMZ has been considered a significant improvement in GBM treatment, it just prolongs the survival time of patients by 2 months on average [2,3,4]. Furthermore, according to research published by CA Cancer J Clin in 2020 [5], many patients with recurrent GBM are not admitted into continuous treatment, which basically indicates the failure of the first surgery. Until now, there is no treatment standard for recurrence and no drugs that extend the survival time. Thus, more effective treatment options to change the poor outcomes of GBM patients are urgently needed.

Researchers have been exploring immunotherapy and precision oncology approaches in recent years. Although some of them have revealed great potential when applied in vitro or in vivo, many of them failed in clinical trials due to unknown reasons. The main factors underlying the high invasiveness and recurrence of GBM are rather complex. There have been some suggestions about paying more attention to the tumor microenvironment, including the extracellular matrix (ECM), communication between glioblastoma stem cells (GSCs; or brain tumor initiating cells, BTICs) and glial cells, etc. [6,7,8,9,10]. Neurotransmitters have long been recognized as prominent signaling molecules of neurogenesis and homeostasis in the CNS [11]. It is found that neurotransmitters could make an overall impact on GBM behaviors through reusing signaling pathways in the normal neural microenvironment, thus, providing a possible way to uncover the mystery of GBM biology [12,13].

By summarizing recent results, we identify ten neurotransmitters as key players in GBM behaviors, which could be classified according to their chemical constitution: (1) acetylcholine (Ach); (2) amino acids, such as glutamate (Glu) and gamma-aminobutyric acid (GABA); (3) biogenic amines, such as dopamine, serotonin (5-HT) and norepinephrine (NE); (4) neuropeptides, such as substance P (SP) and neuropeptide Y (NPY); (5) purines, such as adenosine triphosphate (ATP); (6) gases, such as nitric oxide (NO) [14]. Here, we aim to address three critical areas: (i) we investigate the role of neurotransmitters in the normal neural microenvironment and the GBM microenvironment; (ii) we discuss the key molecular pathways associated with the growth and progression of GBM at the cellular level; (iii) we provide an overview of current knowledge regarding the connections between each neurotransmitter and GBM and focus on the development of therapeutic strategies targeting neurotransmitters.

## 2. Neurotransmitters Function as Necessary Components of Neural Microenvironment and GBM Microenvironment

Traditionally, neurotransmitter signaling is mediated by neurotransmitters released from synapses (‘phasic activation’). However, neurotransmitters diffused from synapses or secreted from non-synapses can cause the activation of target cells as well (‘tonic activation’) [15]. In other words, in addition to being produced by the autonomic nervous system, neurotransmitters can also be released from immunocytes and tumor cells [11]. Meanwhile, numerous non-neurons, such as astrocytes and tumor cells, could react to neurotransmitters via expressing receptors [15,16,17].

It has been demonstrated that neurotransmitters have an impact on adult neurogenesis [15]. Although neurogenesis mainly occurs in the embryonic and early postnatal brain, neural stem cells (NSCs) and neural progenitor cells (NPCs) from the subventricular zone (SVZ) in the lateral ventricles and the subgranular zone (SGZ) in the dentate gyrus of the hippocampus are able to produce new neurons and glial cells in adult brains [18,19]. In these areas, a specific microenvironment, composed of brain microvessels surrounded by neurons and non-neurons, supports the preservation of some characteristics that are recognized as stem cell properties, such as the proliferative and self-renewal capacity of cells. The non-neuron components include NSCs, NPCs, ependymal cells, astrocytes, microglia, macrophages and the ECM [12,19].

By an autocrine or paracrine action, neurotransmitters establish a close link with neurogenesis, especially in the cell proliferation, migration, survival and differentiation of NSCs/NPCs. For example, recent studies have identified the connection between mAChRs and adult hippocampal neurogenesis both in vitro [20] and in vivo [21]. Meanwhile, ACh-nAChRs contribute to the postnatal reactivation of NSCs and adult neurogenesis by maintaining the survival of immature neurons and promoting the proliferation of NSCs in SVZ and SGZ [22]. In the adult brain, the activation of glutamate receptors (GluRs) in NPCs correlates with neurogenesis via inducing cell proliferation, migration, survival and cell differentiation of NPCs [23]. NPY in SVZ and dentate gyrus acts as a regulator of the proliferation, differentiation and migration of NSCs [24].

Furthermore, neurotransmitters act as a universal currency for intercellular communication to connect different types of cells in the neural microenvironment. For instance, GABA mediates the intercellular communication between NPCs in SVZ, playing a role in controlling the migration and cell proliferation of NPCs [25]. ATP has been well-known as energy currency and is involved in various kinds of physiological activities. In general, ATP is released from synaptic vesicles and is able to communicate with glia around the synapse. Glial cells can also release ATP to neurons. In this process, purinergic signaling plays the role of key position in communication between neurons and glial cells [26] (further information in Table 1).

Similar to the neural microenvironment, neurotransmitters act as necessary components of the GBM microenvironment as they infiltrate tumor tissue and exert a regulatory role in the proliferation, migration, apoptosis, autophagy, metabolism, survival, differentiation and angiogenesis of GBM. It is not surprising that GBM cells may retain responsiveness to neurotransmitters [85]. On the one hand, since glioma cells share some markers and molecular characteristics with NSCs [86], it is recognized that glioma may arise from NSCs in SVZ [87]. However, there are also studies suggesting that glioma could originate from astrocytes or oligodendrocyte precursor cells (OPCs) [88], indicating the need for further research. On the other hand, similar to the neurogenesis microenvironment induced by NSCs, a specific microenvironment composed of tumor cells, neurons, stem cells, fibroblasts and vascular and immune cells has been identified in GBM [19,89]. The expression of neurotransmitter receptors has been found on all kinds of cells in the GBM microenvironment [11].

Growing evidence has confirmed the presence of GSCs, a cell subset of tumor cells that exhibit stem cell-like characteristics and express NSC markers [90]. Notably, GSCs have been considered a crucial factor in the invasion, recurrence and chemo-/radio-resistance of GBM [12,91,92]. Unlike NSCs, GSCs are characterized by uncontrolled proliferation and the ability to form a heterogeneous tumor mass through differentiating to heterogeneous cell populations [12,13,93]. Furthermore, GSCs have great potential to develop into TMZ-resistant GBM (TR-GBM) cells, especially under continuous low-dose TMZ stimulation. What should be noticed is that this kind of epigenetic plasticity is reversible, unlike the usual differentiation. It is, therefore, critical to check GSCs’ status before targeting them, to determine whether they are quiescent or proliferative [94]. In addition, the communication between GSCs and the GBM cells also contributes to drug resistance. For instance, GSC-derived PD-L1-containing exosomes have been found to activate Adenosine 5‘-monophosphate (AMP)-activated protein kinase/autophagy-related protein-1 homolog(AMPK/ULK1) pathway, suppress apoptosis and promote protective autophagy, thus, increasing TMZ-resistance in GBM [95]. Through regulating neurotransmitter-associated activities, GSCs will be silenced to some extent and, thus, reducing GBM invasion, recurrence and chemo-/radio-resistance.

## 3. Neurotransmitters and Their Activation on Intracellular Signaling of GBM

The Cancer Genome Atlas (TCGA) GBM dataset shows the transcriptomic and genomic changes of GBM [96]. Researchers have been exploring the mechanisms of molecular alterations in order to determine core signaling pathways that drive GBM pathogenesis. Researchers identified critical genomic mutations responsible for GBM progression: the PI3K pathway (PIK3CA, PIK3R1, PTEN, EGFR, PDGFRA, and NF1), the Rb pathway (CDK4, CDK6, CCND2, CDKN2A/B, and RB1) and the p53 pathway (MDM2, MDM4, and TP53) [97,98].

Many studies have suggested the role of neurotransmitters in intracellular molecular pathways. Of note, these molecular pathways are hijacked by GBM as well. In general, the most common intracellular signals involved are receptor tyrosine kinase (RTK) pathways [73,99], G protein-coupled receptor (GPCR) pathways [100], apoptosis and autophagy signaling pathways [101] (Figure 1).

The neurotransmitters in our studies all have receptors on cell membranes except NO. These receptors can be classified as ionotropic receptors or GPCRs. Ionotropic receptors complete transmembrane signal transduction via ion flux (including Na^+^, Ca^2+^ and Cl^−^), while GPCRs do that by initiating signal cascades mediated by signaling proteins [14]. In addition, we identify a signal network between Calcium (Ca^2+^), RTKs and GPCRs. ACh, Glu, GABA and ATP secretion activate Ca^2+^ flux via ionotropic receptors, serving as a Ca^2+^ entry from the extracellular region to the intracellular region [102]. Moreover, RTKs and GPCRs are also involved in Ca^2+^ release from the endoplasmic reticulum (ER) [103]. In summary, Ca^2+^ has been recognized as an initial impulse and critical modulator of GBM signaling involved in GBM invasion and migration. Research has demonstrated the significant role of GPCR-RTK signaling and Ca^2+^ signaling in non-canonical modes of RTK activation in GBM [104].

RTK signaling pathways are activated by RTKs, including epidermal growth factor receptors (EGFRs), vascular endothelial growth factor receptors (VEGFRs), hepatocyte growth factor receptors (HGFRs) and so on. Previous investigations found that abnormal activation of EGFRs is widely seen in tumor change, related to drug resistance and poor clinical outcome in patients with GBM. Unfortunately, glioblastoma cells are insensitive to EGFR tyrosine kinase inhibitors (EGFR-TKIs), creating a dilemma for drug treatment [105,106,107,108]. However, there is some interesting research exploring the underlying reasons. For instance, EGFR amplification has been identified as an independent predictor of longer survival in older patients diagnosed with GBM [109]. Furthermore, Furnari, F. B. et al. summarized the molecular mechanisms of EGFR amplification and mutations in GBM, and most importantly, revealed the intrinsic EGFR heterogeneity in GBM [110]. These findings deepen the current understanding of EGFR-targeted therapy, which is of great significance in drug design. RTK signaling pathways get involved in the modulation of cell growth, differentiation and reproduction [104,111]. There are two critical pathways altered in GBM, the Ras/MAPK (Ras protein/mitogen-activated protein kinase) and PI3K/AKT/PTEN (phosphatidylinositol 3-hydroxy kinase/protein kinase B/PTEN Phosphohydrolase), resulting in oncogenic changes that ultimately lead to uncontrolled invasion and drug resistance. The PI3K/Akt/PTEN pathway is constitutively active due to PTEN alterations, PIK3CA mutations or Akt amplification in GBM [112]. PTEN has been recognized as one of the chromosome 10 tumor suppressor genes in GBM progression [113]. The relationship between PI3KCA and PTEN has been discussed back in 2005. PIK3CA mutation can encode a protein that antagonizes PTEN in the PI3K/Akt pathway [114]. Researchers have summarized the potential mechanisms for resistance, mainly focusing on targeting RTKs [104,115].

Of note, the mammalian target of rapamycin (mTOR) pathway plays a vital role in autophagy induction [116]. For instance, mTOR can be activated through the protein kinase B (PKB/AKT) or mitogen-activated protein kinase (MAPK) pathway, thereby inhibiting cell autophagy [117]. In addition, studies have revealed that PI3K-PTEN-AKT–mTOR signaling emerges as a type of fundamental signaling pathway in the regulation of cell activities, such as cell proliferation and cell death [111]. Taking ONC201 as an example, as a dopamine receptor D2 (DRD2) antagonist, it inactivates protein kinase B/extracellular-regulated protein kinases (AKT/ERK) signaling in tumor cells [118]. Moreover, LY341495, which is a mGluR2/3 antagonist, inhibits the ERK1/2 and PI3K/AKT/mTOR pathways, thus, reducing cell growth [119].

Further, the chief regulator of cell survival is the NF-κB (nuclear factor-kB) pathway, which has been identified to be activated in over 50% of all cancers, including GBM [120]. It is known that GBM is featured with high heterogeneity [93], drug resistance and a high level of NF-kB [79]. As a transcription factor, NF-kB has been shown to get involved in tumorigenesis, tumor growth and tumor response to drugs [121]. Therefore, NF-kB followed by signaling pathways (such as signal transducer and activator of transcription 3, STAT3) has been considered the main driver in GBM [122]. Furthermore, research has shown that NF-κB is relative to all kinds of different neurotransmitters directly or indirectly, including NO, Glu and so on, thus, occupying a significant position in adjuvant therapy [123,124].

The pathways that mediate apoptosis are divided into external pathways and internal pathways. The external pathway activates pro-caspase-8 and pro-caspase-10 through the connection of pro-apoptotic ligands and death receptors extracellularly, thus, activating caspase-3 to complete cell apoptosis. Instead, the intrinsic pathway is caused by mitochondrial alterations and implicated in intracellular signaling mediated by non-receptors [125]. Strikingly, neurotransmitters have been shown to be involved in the activation of pro-caspase-8/9/10/3 and promote cell apoptosis in GBM cell lines [46,126,127].

The interactions with the ECM are also one of the neurobiological roots maintained by GBM [85]. Neurotransmitter signaling, including Ach [27], Glu [47], NE [62,128] and SP [76], plays a prominent role in the regulation of invasion behavior of GBM via acting on matrix metalloproteinases (MMPs) (Figure 2). MMPs, which are a group of proteolytic enzymes containing Zn^2+^ secreted by cancer cells, have the ability for dissolving various kinds of ECM that includes fibronectin, Fibrinogen, chondroitin sulfate proteoglycans (CSPGs), laminin, heparan sulfate proteoglycans (HSPGs), collagens and tenascin-C [129,130]. Furthermore, studies have revealed that GBM cells prefer to migrate to a more rigid microenvironment, resulting in the impairment of tissue morphology and promoting tumor growth and development [130]. The hardness of the microenvironment is determined by the composition of ECM: both the enhanced matrix crosslink in the ECM and the anchoring of integrin-focal adhesion complexes (FAK) between cells and the ECM have provided the desired properties to allow tumor invasion [49].

Overall, at the cellular level, neurotransmitters in GBM retain the ability to transform ligand binding on the cell membrane into intracellular signaling. Elucidating common pathways associated with neurotransmitters helps in understanding the next part and facilitates the better use of neurotransmitters as a tool for treating tumors.

## 4. The Influence of Neurotransmitters on GBM and Potential Therapeutic Targets

Since the potent infiltrative capability of GBM makes it impossible to cure the tumor only by surgical resection, radiotherapy and chemotherapy after surgery are crucial [131]. As mentioned before, current drugs for chemotherapy are insufficient and have low-effectiveness. In this regard, the growing understanding of how neurotransmitters influence the initiation and advance of GBM has led to the development and evaluation of potential drugs targeting neurotransmitters and their receptors. In the following text, we review the existing advancement, provide a list of related experiments in Table 2 and display the targets of neurotransmitter-related drugs in Figure 3, attempting to provide a reference for future drug selection. (Due to the downregulation of GABA receptors in GBM and the lack of experiments to prove the role of Y2 receptors in GBM, we did not show these two neurotransmitters (GABA and NPY) and their receptors in Figure 3. Detailed discussion can be found in Section 4.3 and Section 4.8.1).

Of note, drugs inhibit tumor growth mainly by interfering in the balance of cell proliferation, apoptosis and autophagy, thus, serving active participation in the response of tumor cells. For example, the mutation of mitochondria DNA (mt-DNA) causes the functional disorder of mitochondria and the activation of intracellular signaling pathways, including AKT, Ras and ERK pathways, as well as the activation of hypoxia-inducible factor 1α (HIF-1α) and NF-κB, thus, regulating the expression of nuclear genes associated with drug resistance [132]. Furthermore, research has found that fluoxetine induces transmembrane Ca^2+^ influx and causes mitochondrial Ca^2+^ overload, thereby triggering apoptosis [133].

**Table 2 cancers-14-03970-t002:** Potential effects of drugs targeting neurotransmitters in treating GBM.

Drugs	Targets	Approved by FDA?	Effects	Type of Study	References
Basic Studies					
Arecaidine propargyl ester (APE)	M2 mAChR agonist	NO	(i) suppress cell cycle progression via inhibiting mitotic catastrophe both in GBM lines and in GSC lines, (ii) induce oxidative stress and cell apoptosis in p53 wild-type GBM lines, (iii) counteract the adaptive responses to hypoxia conditions in GB8	in vitro (GBM line U251MG and GBM line U87MG, GSC line GB7 and GSC line GB8)	[134,135]
Iper-8-naphthalimide(N-8-Iper)	M2 mAChR agonist	NO	(i) cytotoxic effects in GSCs, (ii) a cell growth inhibition and severe apoptosis by inducing DNA damage at lower doses than APE, (iii) exhibit antinociception mediated by muscarinic receptors without relevant cholinergic side effects	in vitro (GSC line GB7); in vitro (GSC line GB7 and GSC line G166)	[31,136]
Iper-6-phthalimide(P-6-Iper)	M2 mAChR agonist	NO	no appreciable effects on GSCs growth	in vitro (GSC line GB7)	[31]
Atracurium Besylate	nAChR antagonist	YES	(i) induce astroglial effectively but not neuronal differentiation of GSCs, (ii) inhibit the replication of the patient-derived GSC line	in vitro (GSC line HSR-GBM1, GSC line HSR040622 and GSC line HSR040821)	[32]
StN-2(MG624)	α7 and α9 nAChR antagonist	NO	(i) suppress the cell proliferation in a dose-dependent manner, (ii) decrease ATP production after 72 h	in vitro (GBM cell line U87)	[137]
StN-4	α7 and α9/10 nAChR antagonist (silent agonist)	NO	(i) a more pronounced inhibitory effect on the viability of human U87MG cells as compared to MG624, (ii) decrease the proliferation of U87MG cell, as well as the level of pAKT and oxphos ATP, (iii) induce G0/G1 cell cycle arrest and cell apoptosis, (iv) decrease the mitochondrial and cytoplasmic ATP production after 1 h, (v) more active in GBM than in normal astrocytes of mice	in vitro (human U87MG and GBM5 cells)	[138]
StN-8	α7 and α9/10 nAChR antagonist	NO	(i) a more pronounced inhibitory effect on viability of human U87MG cells and a less inhibitory effect on viability of mice astrocytes as compared to MG624, (ii) decrease the proliferation of U87MG cell, as well as the level of pAKT and oxphos ATP, (iii) decrease the mitochondrial and cytoplasmic ATP production after 1h, (iv) more active in GBM than in normal astrocytes of mice	in vitro (human U87MG and GBM5 cells)	[138]
3-(2,4-dimethoxybenzylidene) anabaseine (GTS-21)	α7 nAChR antagonist	NO	inhibit cell proliferation in a dose- and time-dependent manner in an α7-nAChR-dependent/α7-nAChR-independent manner	in vitro (GBM cell line A172, U87 and G28, and patient-derived glioblastoma cells)	[35]
Kynurenic acid (KYNA)	α7 nAChR and all iGluR antagonist	NO	(i) suppress the cell proliferation, (ii) enhance the inhibition of MK801 and GYKI 52466 on cell proliferation, (iii) elicit a prominent inhibitory effect on glioma cell motility at low dose	in vitro (GBM cell line T98G)	[139]
Amb4269951	Choline Transporter-Like Protein 1(CTL1) inhibitor	NO	(i) choline uptake and cell viability inhibition in vitro, (ii) increase caspase-3/7 activity in vitro, (iii) promote cell apoptotic through the ceramide-induced inhibition of survivin expression in vitro, (iv) induce antitumor effects without loss of weight in xenograft mice by suppressing the growth of tumor	in vitro (GBM cell line U251MG) and in vivo (mice xenograft models)	[126]
Dizocilpine (MK801)	NMDAR (N-methyl-D-aspartate receptor) antagonist	NO	(i) decrease cell invasion, (ii) suppress the expression of GluN2 and GluA1 subunits upregulated by NMDA stimulation	in vitro (GBM cell line LN18, GBM cell line U251MG and patient-derived glioblastoma cells)	[140]
Memantine	NMDAR antagonist	YES	induce cytotoxicity of GBM in a dose-dependent fashion	in vitro (human GBM cell line T98G and U87MG)	[141]
MP1-MP2	NMDAR antagonist	NO	induce significant cell death by apoptosis	in vitro (human GBM cell line U87MG)	[142]
Ifenprodil	NMDAR GluN2B antagonist	YES	more potent than MK801 in inhibiting cell migration and survival, as well as in the sensitivity to radiation	in vitro (cell lines)	[40]
Perampanel (PER)	AMPAR antagonist	YES	(i) significant cell growth inhibition, (ii) suppress Ca^2+^ permeability and induce high levels of cell apoptosis, (iii) upregulate GluR expression in U87 and U138 cell line (i) Glucose uptake attenuation in all glioblastoma cells, (ii) have no induction on cell apoptosis, (iii) reduce exorbitant extracellular glutamate levels	in vitro (GBM cell line U87, U138 and A172; GBM cell line HROG02, HROG05, HROG15 and HROG24)	[143,144]
GYKI 52466	AMPAR GluR1 antagonist	NO	diminish cell death induced by cancer in peritumor brain	in vitro (rat F98 glioma-implanted brain slice cultures)	[145]
Fluoxetine	AMPAR GluR1 antagonist	YES	(i) induce transmembrane Ca^2+^ influx and cause mitochondrial Ca^2+^ overload, thereby triggering apoptosis, (ii) suppress the growth of glioblastomas in mice	in vitro (GBM cell line U87 and Hs683 and rat glioma cell line C6) and in vivo (experimental animal models)	[41]
Riluzole	mGluR1 antagonist	YES	(i) lead to the caspase-dependent cell apoptosis and an accumulation of cells in phase G0/G1 in a dose-dependent fashion, thus, resulting in the reduction of cell activity and the promotion of LDH release, (ii) attenuate cell invasion and migration of GBM, (iii) induce the death of GBM cells through suppressing mGluR1/PI3K/AKT/mTOR signaling, (iv) inhibit GSC cell growth through inhibition of glucose transporter 3 (GLUT3) that associates with a decrease in p-AKT/HIF1 alpha pathway	in vitro (GBM cell line U87, patient-derived glioblastoma cells) and in vivo (U87 cell xenograft model)	[127,146]
BAY36-7620	mGluR1 antagonist	NO	(i) lead to the caspase-dependent cell apoptosis and an accumulation of cells in phase G0/G1 in a dose-dependent fashion, thus, resulting in the reduction of cell activity and the promotion of LDH release, (ii) attenuate cell invasion and migration, (iii) induce the death of GBM cells through suppressing mGluR1/PI3K/AKT/mTOR signaling	in vitro (GBM cell line U87) and in vivo (U87 cell xenograft model)	[127]
CPCCOEt	mGluR1 antagonist	NO	(i) reduce GBM cell activity in a dose-dependent fashion, (ii) inhibit mGluR1 signaling via Gaq pathway and β-arrestin-dependant pathway	in vitro (GBM cell line A172 and U87)	[147]
JNJ16259685	mGluR1 antagonist	NO	(i) reduce GBM cell activity in a dose-dependent fashion, (ii) inhibit mGluR2 signaling via Gaq pathway and β-arrestin-dependant pathway	in vitro (GBM cell line A172 and U87)	[147]
(2S)-α-ethylglutamate (EGlu)	mGluR2/3 antagonist	NO	cell proliferation inhibition in a time-dependent fashion	in vitro (GBM cell line U87)	[119]
MTPG	mGluR2/3 antagonist	NO	cell proliferation inhibition in a time-dependent fashion	in vitro (GBM cell line U87)	[119]
LY341495	mGluR2/3 antagonist	NO	(i) cell growth reduction, (ii) decrease expression of cyclin D1/2, (iii) inhibit the ERK1/2 and PI3K/AKT/mTOR pathways	in vitro (GBM cell line U87)	[119]
Sulfasalazine (SSZ)	xc−cystine/glutamate transporter antagonist	YES	(i) reduce glioma cell growth at high concentrations, (ii) induce endoplasmic reticulum stress and ferroptotic cell death with no influence on cell autophagy, (iii) minor impact on astrocytes and does not affect neuronal viability, (iv) not affect tumor growth progression, but alleviates tumor-related brain edema	in vitro (rat glioma cell F98, human glioma cell U251, primary astrocytes and neurons, and rat organotypic brain slices)	[148]
VU0155041	mGluR4 antagonist	NO	(i) decrease cell activity in a dose- and time-dependent fashion, (ii) block the expression of cyclin D1 induced by SHH and GBM proliferation, (iii) increase TUNEL-positive cells and apoptosis-related proteins activation	in vitro (GBM cell line LN229)	[46]
Quetiapine	DRD2 antagonist	YES	(i) reduce glioma cell self-renewal in vitro, (ii) prolong the survival of glioma-bearing mice, (iii) induce the expression of genes involved in cholesterol biosynthesis through combined treatment	in vitro (patient-derived HK-157, HK-308, HK-374, and HK-382 GBM lines) and in vivo (the GL261 orthotopic mouse models of GBM, and HK-374 patient-derived orthotopic xenografts)	[149]
Perphenazine	DRD2/3 and 5-HT antagonist	NO	(i) reduce centrifugal migration of cells in SVZ, (ii) activate protein phosphatase 2, (iii) reduce nausea	in vivo	[150]
Valerenic acid	HTR5A agonist	NO	(i) induce cytotoxicity via apoptosis/autophagy, (ii) trigger oxidative stress, (iii) suppress proliferative effect and Epithelial-Mesenchymal Transition (EMT), (iv) inhibit cell migration and invasion, (v) suppress xenograft glioblastoma	in vitro (U251 MG human glioblastoma cell lines)	[60]
Brexpiprazole	DRD2 and 5-HT1A agonist, 5HT2A antagonist	YES	increase the sensitivity of GSCs to osimertinib through the downregulation of survivin expression	in vitro (GSCs:A172GS, GS-Y01, GS-NCC01, and GS-Y)	[151,152]
Clenbuterol hydrochloride	β2-AR agonist	NO	upregulate Cx43 expression that promotes cell communication	in vitro (OECs)	[153]
AZ10606120	P2X7R antagonist	NO	(i) decrease GM-CSF mRNA and protein expression, (ii) significantly decrease U251 MG cell proliferation	in vitro (U251 MG cells)	[154,155]
Paris saponin H (PSH)	ARA1 and ARA3 inhibitor	NO	(i) inhibit cell viability, migration and invasion, (ii) induce cell apoptosis, (iii) induce G1 cell cycle arrest via the increase in the expression of p21 and p27, and the decrease in the expression of cyclin D1 and S-phase kinase associated protein 2	in vitro (U251 cells)	[156]
1400W, S-MIU	iNOS inhibitor	NO	inhibit the cell growth of astrocytes and U87 cell line with EGFRvIII expression	in vivo (U87 cell line, rat model)	[157,158]
**Clinical Studies**					
Memantine	NMDAR antagonist	YES	in safe combinations with TMZ in newly diagnosed GBM with a 21-month median survival and a 2-year survival rate of 43%	Phase I, randomized, (newly diagnosed, n = 85)	[159]
Talampanel (LY300164)	AMPAR antagonist	NO	(i) no appreciable effects on unselected recurrent gliomas in talampanel monotherapy, (ii) well-tolerated without significant additional toxicity, (iii) superior survival and lower O(6)-methylguanine-DNA methyltransferase methylation of newly diagnosed glioblastoma patients receiving RT+TMZ and talampanel than only RT+TMZ treatment	Phase II, single-agent (recurrent, n = 22) Phase II, single-arm (newly diagnosed, n = 72) Phase II, randomized, (newly diagnosed, n = 365)	[160,161,162]
ONC201	DRD2 antagonist	NO	(i) Ca²⁺ flux is involved in the anti-tumor cells activities of D2R antagonist, (ii) reduced self-renewal, clonogenicity and cell viability in secondary and tertiary glioma spheres, (iii) ONC201 combined with radiation prolonged survival in mouse models, (iv) gene expression alteration associated with GBM plasticity, quiescent populations and GBM stem cells. (i) well-tolerated with no drug-attributed toxicity, (ii) higher concentration than research conducted in glioblastoma neurosphere cultures, (iii) one patient with subcentimeter, multifocal, recurrent glioblastoma exhibits a complete regression of her enhancing lesions	in vitro (U87MG, U251MG, U373MG, Hs683, SF-295, A172 and LN-18 cell lines) and in vivo (patient-derived GBM cell lines and orthotopic xenograft mouse models) Phase II, non-Randomized, (recurrent, n = 140)	[163,164,165]
Chlorpromazine	DRD2 antagonist	YES	(i) reduce GBM cell viability, (ii) induce cell cycle alterations and cause hyperdiploidy in GBM cells, (iii) reduce GBM cell cloning efficiency, (iv) downregulate stemness gene expression in GBM cells, (v) synergize with TMZ in reducing the viability and cloning efficiency of GBM cells and inducing cell death	in vitro (cell lines T98G, U-251 MG and U-87 MG) Phase II, recruiting, (newly diagnosed, n = 41)	[166]
Escitalopram oxalate	SSRI (selective serotonin reuptake inhibitors)	YES	(i) inhibit cell reproduction and invasion in U-87MG cell line, (ii) reduce the expression of cell cycle inhibitors, including Skp2, P57, P21 and P27, (iii) induce apoptotic cascades in U-87MG cell line and autophagy in GBM8401 cell line	in vitro (U87MG cells GBM8401 cells) Phase II, Randomized, (newly diagnosed, n = 100)	[167]

### 4.1. Acetylcholine (ACh): Regulates GBM Proliferation, Survival and Invasion

GBM may be able to synthesize ACh, which can be used as an autocrine and paracrine signaling to modulate tumor proliferation, survival and invasion [27]. The activation of AChRs, for instance, promotes the invasion of GBM via an MMP-9 activity mediated by intracellular Ca^2+^ increase [27]. Recent studies have indicated that M2 mAChRs exhibit inhibitory effects on the cell proliferation and survival in GBM cell lines and GSC lines. Specifically, M2 mAChRs induce apoptosis and cell cycle arrest of GBM cells, whereas they hamper the cell cycle and inhibit cell proliferation mediated by Notch-1/EGFR pathways in GSC lines [28,29,30,31]. In an elegant study, M3 mAChRs (CHRM3) appear to be implicated in the decrease of patient survival [27]. As for nAChRs, the expression of α1nAChRs (encoded by CHRNA1) and α9nAChRs (encoded by CHRNA9) is elevated and is related to reduced patient survival [27]. It is found that these two receptors correlate with the maintenance of self-renewal of GSCs in an ex-vivo experiment [32]. In GBM cell lines, the activation of α7nAChRs and α9nAChRs has shown the inhibition of cell apoptosis (via EGFR/AKT pathways) and promotion of cell proliferation (via EGFR/ERK pathways) [33,34]. What is more, the increase in α7nAChRs in GBM cells, vessel endothelium and tumor-associated macrophages of GBM have been identified, inducing the inhibition of cell proliferation [35]. The activation of AChRs (mainly α7 nAChRs) on astrocytes and microglia may also prevent excessive neuroinflammation, which causes damage to neurons and glial cells [36]. Up to now, few studies have linked the “brain cholinergic anti-inflammatory pathway” with GBM, which deserves further research.

The various relationships between GBM and AChRs, especially M2 mAChRs and α7/α9 nAChRs, support AChRs as a promising target for developing innovative antitumor therapies. N-8-Iper is the most potent agonist among the selective M2 mAChR agonists in Table 2, with approximately 100 times the potency of ACh [31]. The upregulation of some certain receptors in tumor cells is conducive to the tumor-specific action of selective ligands, without toxicity to normal cells. Since many experiments and public databases (such as accession number GSE16805) have proved that the expression of α9 nAChRs is higher in GBM [27,32,34] (no expression in normal mammalian brain [168,169,170], α9 nAChRs may be a powerful promising target. Atracurium Besylate (nAChR antagonist) [32], MG624 (α7 and α9 nAChR antagonist) [137] and StN-4/StN-8 (α7 and α9/10 nAChR antagonist) [138] have shown significant inhibitory effect on the viability of GBM/GSC cell lines. However, it has been reported that some of their anti-tumoral effects are nAChR-independent, which increases the difficulty of elaborating their mechanisms in GBM and needs additional investigations [35,138]. Strikingly, targeted drugs for AChRs combined with chemotherapeutic agents represent a new prospective treatment for GBM. For instance, the combination of N-8-Iper and doxorubicin, cisplatin or temozolomide shows a stronger impairment of the growth of GSC lines than that with the single treatment. This may be related to the decrease in the ATP-binding cassette (ABC) efflux pumps mediated by the M2 receptor, reducing the excretion of chemotherapeutic drugs [136].

It is shown that the choline metabolism increases in many tumors and the transporting system of choline, which has been recognized as an important rate-limiting step, which could be a promising target in GBM therapy [27]. Although the potential effect of Amb4269951, a Choline Transporter-Like Protein 1(CTL1) inhibitor, has been distinguished in vitro and in vivo [126], little is known about the choline transport system in GBM [171].

Notably, the blood–brain barrier (BBB) that prevents brain tumors from being infiltrated by drugs is an obstruction to be reckoned with for drug consideration. Studies have identified that nAChRs could be implicated in drug delivery through receptor-mediated transcytosis (RMT), providing a promising way for the intracranial transportation of drugs that avoids the BBB [172]. Taken together, future research should further identify potent selective AChR agonists/antagonists to develop targeted drug-screening strategies for GBM treatment.

### 4.2. Glutamate (Glu): Modulates GBM Growth and Progression, as Well as Increases Radiosensitivity

Recent studies have demonstrated the regulation of both synaptic and non-synaptic glutamate on embryonic neurogenesis and the migration of neurons and astrocytes. To our knowledge, these physiological processes are absent in adult brains, but could be found in glioma [39].

Multiple effects of GluRs have been found in GBM: (i) the phosphorylation of transcription factor CREB followed by NMDAR activation is responsible for the promotion of GBM survival and migration, as well as the increase in radiosensitivity through inhibiting the repairment of the DNA double-strand break (DSB) [40]; (ii) AMPARs mediate the Ca^2+^-dependent activation of the AKT/PKB signaling pathway, stimulating tumor growth and invasion [41,42,43,44]; (iii) different subtypes of mGluRs induce opposite effects on GBM. Studies have exhibited that more expression of mGluR3 implicates a higher malignancy of tumors and a higher fatality [45], consistent with the result of D’Alessandro, G., et al., who reported the inhibition of GBM proliferation caused by the block of mGluR2/3 [42]. Instead, mGluR4 may inhibit cell proliferation and induce apoptosis by (i) decreasing cyclin D1 expression and increasing pro-caspase-8/9/3, (ii) or regulating the expression of Bcl-2/Bax, (iii) or reducing Gli-1 expression that emerges as a transcription factor of Sonic Hedgehog (SHH) signaling [46]. Significantly, both NMDARs and APAMRs show an influence on the extracellular matrix. For one thing, NMDARs are implicated in the activation of MMP2 [47]. For another, the high expression of GluR1 may promote the expression of integrins (β1 integrins mainly) on the glioma cell membrane, which, in turn, increases the adhesion to the ECM and drives migration along perivascular and subpial spaces [48].

Accumulating evidence suggests that glutamate is an important regulator of cellular activities in both the neural and GBM microenvironment: high concentrations of glutamate in extracellular space leads to not only the excitotoxic cell death of neurons, but also the promotion of tumor growth, migration and invasion [44,49,50]. As a result, it is necessary to elucidate how the regulation of glutamate levels performs in the neural and GBM microenvironments.

Currently, the active participation of neurons, astrocytes and microglia has been identified. It is well-established that tripartite synapses composed of synaptic neurons and astrocytes produce the glutamate cycle. In the tripartite synapse, glutamate released to the synaptic cleft not only acts on postsynaptic neurons, but also diffuses around astrocytes, rendering an activation of GluRs on astrocytes and a glutamate uptake. Subsequently, astrocytes secrete glutamate to the presynaptic neuron, resulting in the glutamate cycle [173,174]. It is shown that glutamate uptake and release also occur in microglial cells [175]. Overall, glial cells uptake glutamate through glutamate transporters (EAAT) and glutamine synthetase (GS), and glutamate release may be related to the activation of glutamate-cystine antiporters, Best1 channels and Cx43 hemichannels [175]. Furthermore, microglia can also inhibit the glutamate release of glioma cells via transferring miR-124 (a tumor suppressor) to astrocytes and tumor cells by releasing Small Extracellular Vesicles (SEVs) [176]. Compared to the normal neural microenvironment, the release and conduction process of glutamate in the GBM microenvironment are altered and cause a high level of extracellular glutamate [49], which may be mediated by the following mechanisms: (i) a 100-fold excess spontaneous secretion of glutamate compared to normal levels (>100 µM), induced by the upregulation of the cystine/glutamate antiporter xc-(System xc-, SXC) and an instantaneous intracellular Ca^2+^ increase caused by P2X7, which is activated by high levels of ATP in the extracellular matrix; (ii) reduced reuptake of glutamate due to the decrease in excitatory amino acid transporter (EAAT2) expression; (iii) the failure of astrocytes to clear glutamate in the extracellular milieu [40,49,177,178].

The inhibitory effects of iGluR antagonists and mGluR blockers on GBM growth have been reported both in vivo experiments and in vitro experiments (Table 2). Indeed, since the interference of mGluRs in rapid excitatory impulse transmission seems much smaller than that of iGluRs, the mGluR might be a more appropriate target that exhibits less untoward reaction in patients [179]. Furthermore, the silence of mGluR1 genes has been shown to evidently reduce glioma proliferation [147].

Remarkably, diverse derivatives of memantine could be potential antitumor agents for GBM therapy as well. For example, the synthesized memantine-derived compounds MP1-10, especially MP1 and MP2, have been improved to suppress cell proliferation of T98G and U87-MG in a dose-dependent fashion. Nevertheless, further biological studies show that only MP1 exhibit appropriate solubility and enzyme stability in gastric fluids and intestinal secretions, which is appropriate for further in vivo pharmacokinetic research [142]. In addition, nitromemantine, which is the second-generation memantine derivatives, not only attenuates NMDAR signaling directly, but also carries NO to the regions with overactive NMDAR, causing high concentrations of NO and superior tumoricidal effects than memantine. However, further verifications on animal models are needed [177].

Combinatorial treatment with TMZ and other drugs, such as riluzole [180], Kynurenic acid (KYNA) [139], perampanel [143] and talampanel [49], has been a promising strategy for GBM therapies as it shows conspicuous advantages in decreasing chemoresistance, enhancing radiosensitivity and improving the survival of patients. Further study also provides a new perspective on medicine combination. For example, combinatorial treatment with fasudil and glutamate exhibits more significant inhibitory effects on cell activity and more synergist effects on lactic dehydrogenase-C4 (LDH) levels than fasudil monotherapy. Fasudil is a Rho-associated coiled-coil containing a kinase (ROCK) inhibitor that has been utilized as a common drug for clinical treatment of subarachnoid hemorrhage (SAH). In an experiment conducted on primary human GBM cells, fasudil has been found to elevate the sensitivity of GBM cells to glutamate excitotoxicity through increasing NMDAR GluN2B expression [181]. Nevertheless, whether this combination treatment is equally effective in animal models remains unknown.

Sulfasalazine (SSZ/SAS), emerged as an SXC inhibitor correlating with the release of glutamate and the intracellular transportation of cysteine, resulting in the decreased production of glutathione, thus, impairing the GBM tolerance to oxidative stress [148]. However, although SSZ has exhibited a good effect without any toxicity in experiment rodents, serious side effects on patients with GBM have been shown in two clinical trials, characterized by significant toxicity for neuronal function, bone-marrow and the hematological system (mainly leukopenia and neutropenia) [182]. Indeed, the increase in extracellular glutamate levels and synaptic damage in rat hippocampal cultures given SXC inhibitors (including sorafenib and erastin) have been identified [183]. Additionally, propofol, involved in the regulation of Ca^2+^ permeable AMPA receptor (CPAR)-SXC pathways, could suppress the proliferation, invasion and migration of C6 glioma cells significantly [184]. More evidence of propofol’s validity in GBM needs to be found in the future. To sum up, current knowledge of the role of glutamate in tumor proliferation, survival, migration, invasion and radiosensitivity advances our understanding of the biology and therapies of GBM. Selective ligands of GluRs and the xc−cystine/glutamate transporter that have reliable safety, high bioactivation in vivo and BBB penetrability may be potential therapeutic strategies for GBM.

### 4.3. Gamma-Aminobutyric Acid (GABA): Inhibits GBM Growth

The activation of GABA_A_R in low-grade glioma could inhibit cell growth by depolarizing the membrane through increasing intracellular Cl^−^ caused by Na^+^–K^+^–2Cl^−^ co-transporter NKCC1. Nevertheless, the decreased numbers of mRNA encoding GABAR subunits and loss of GABA_A_R in GBM may indicate the relationship between the number of functional GABA_A_R and the grade of the glioma [12,52,53]. It also means that GBM may counteract the attenuation of GABA on cell proliferation by decreasing the expression of GABA_A_R [54]. Notably, there are conflicting findings regarding whether GABA can decrease the proliferation of GSCs. Recently, the functional expression of GABA_A_R and a significant increase in the number of tumor cells while applying bicuculline, a GABA_A_R antagonist, have been identified in tumor cells from GBM patients. It is also reported that GABA selectively acts on cells with stem cell-like properties, maintaining the quiescence of GSCs and allowing GSCs to enter cell cycles again after surgical resection of tumor bulks [185]. On the contrary, El-Habr’s studies show that GABA has no inhibitory effects on GSCs [186].

Although GABA may inhibit GBM proliferation, its existence has not been confirmed in experiments so far [187]. The human glioblastoma stem cell line, U3047MG, has been recommended by Babateen, O., et al. as the model that may be suitable for exploring the potential effects of GABA on GBM in the future, due to its abundant expression of GABAR subunits and some functional channels [53].

Because of the lack of GABAR in GBM and its inability to respond to agonists/antagonists, researchers have recently focused on RNA editing and GABA metabolism. RNA editing refers to the process of post-transcription alteration of the RNA sequence via base switching/insertion/deletion, playing a distinguished role in maintaining tissue homeostasis. Recent studies have revealed that the reduction of GABRA3 editing, which is implicated in the editing of α3-GABAR, would promote tumor invasion [188]. Identifying altered RNA editing events in GBM may help to develop alternative gene-therapy options.

GABA, associated with the synthesis of succinate via activating GABA transaminase and succinic semialdehyde dehydrogenase (SSADH), has been recognized as an essential substrate of the TCA cycle and a positive regulator of GBM proliferation [189]. It is likely that GABA oxidation and SSADH activity establish new ways to target GBM [189]. However, specific SSADH inhibitors have not been detected yet. Moreover, the level of GHB (the by-product of GABA catabolism), which could be elevated by the reduction in SSADH expression or the compensation of GHB, is implicated in transforming the glioma from a highly malignant state into a phenotype that is more differentiated and less aggressive [186]. Up to now, GHB has been exploited for multiple pathological symptoms, such as narcolepsy and alcohol withdrawal. Significantly, studies have identified its good properties to pass through the BBB, indicating that it could be an additional therapeutic target in GBM [186]. To date, the GABA metabolism may be a more potent target than GABAR when discovering novel therapies for GBM.

### 4.4. Dopamine: Inhibits GBM Invasiveness and Migration

In general, there appear to be two main mechanisms in dopamine treatment for GBM: inhibiting tumor angiogenesis through the VEGF pathway and inducing cell autophagy mediated by the mTOR pathway or via the Beclin1 (BECN1)-dependent pathway [190,191,192]. Studies have suggested that dopamine inhibits tumor migration by reducing stress-mediated angiogenesis [58]. Surprisingly, the mechanism of how dopamine reduces angiogenesis is still under investigation, which may be a huge breakthrough [193,194,195]. Immune alterations are mainly responsible for stress-induced tumor invasion. The concept of tumor vasculature normalization was put forward by Jain et al. in 2005 [196]. Tumor associated-macrophage (TAM) polarization had been suggested as a potential target for inducing tumor vasculature normalization [197]. Research conducted on C6 glioma-bearing rats [198] proved that low dopamine modulates TAM polarization, induces vascular normalization and, thus, suppresses glioma growth. Dopamine reprograms M2-polarized macrophages by downregulating the VEGF/VEGFR2 signaling, thus, switching M2-polarized macrophages to an M1 phenotype. Furthermore, dopamine has been shown to modulate the microglial cell inflammatory response induced by IL-1b, IL-6 and inducible nitric oxide synthase (iNOS) [199,200,201]. Dopamine acting on immune-competent cells has revealed great influence on glioma development and progression. On the one hand, dopamine inhibits tumor angiogenesis. On the other hand, dopamine communicates with the immune system and participates in inflammatory activities. In addition, dopamine has been suggested as an important modulator in autophagy and apoptosis [57,202].

Moreover, DRD2 antagonists have great potential in treating GBM. ONC201 appears to act as a selective DRD2 antagonist. It may not be as strong as other DRD2 antagonists, but it performs better when it comes to clinical use for its wide anticancer effect [118]. DRD5 modulates DRD2 signaling negatively and, thus, weakens the ONC201 drug effect [203]. Moreover, ONC201 enhances radiation and the TMZ effect in GBM cell lines [163]. Chlorpromazine synergizes with TMZ in reducing the viability and cloning efficiency of GBM cells and inducing cell death [166]. A previously published article has shown that perphenazine is excellent in blocking DRD3. It can inhibit SVZ cell growth which is supported by DRD3 activation [204].

Surprisingly, DRD2 and DRD3 agonists were demonstrated to induce autophagy through the BECN1-dependent signaling pathway. This research applies dopamine as an important regulator in autophagy [192]. As mentioned before, cell proliferation is controlled by mTOR and its inactivation induces autophagy. Significantly, an uncontrolled proliferation that results from a decline in apoptosis is responsible for radiotherapy resistance to GBM treatment. Additionally, clinical use of rapamycin can upregulate drug sensitivity in glioblastoma for its inhibition of mTOR signaling [191]. In conclusion, dopamine is promising for further research.

### 4.5. Serotonin (5-HT): Involved in Cell Apoptosis, Autophagy and Chemotherapy Reinforcement

Serotonin activates multiple subtypes of receptors and induces different kinds of responses, as well as GBM heterogeneity [93], thus, requiring further research [190]. For example, current understanding of 5-HT-targeted drugs mostly concentrates on HTR5A and HTR7A. HTR5A was found downregulated in GBM compared to low-grade gliomas including astrocytoma, oligoastrocytoma and oligodendroglioma [60]. It was well-established that valerenic acid, a potential agonist of HTR5A, could trigger autophagy and apoptosis and, thus, induce the death of GBM cells [60]. In U373 GBM cells, interleukin-6 can be induced by serotonin activating on 5-HT7 receptors [205]. In turn, interleukin-6 contributes to supporting GBM progression via the autocrine pathway [206]. Generally, targeting 5-HT receptors is a type of potential way of treating GBM, functioning as reinforcement to chemotherapy [207,208,209].

GSCs represent one of the mechanisms responsible for the high chemoresistance of GBM that is associated with a poor prognosis [95]. In this regard, co-treatment involving 5-HT may be a potent approach to improve chemosensitivity of GSCs. For instance, brexpiprazole is able to amplify GBM response to TMZ by regulating serotonin and dopamine. Furthermore, the combination of brafiprazole and orcetinib is more effective in inhibiting GBM progression by downregulating the expression of surviving, and no significant side effect was confirmed [151].

Researchers have indicated that selective serotonin reuptake inhibitors (SSRIs) are toxic to some tumors and own a potential drug effect on GBM. SSRIs are applied in depression treatment at first and they have surprising outcomes in particular patients who suffer from depression caused by GBM. Following studies discovered that SSRIs suppress GBM cell proliferation in different ways. What is worth mentioning is that some FDA-approved antidepressants already reached favorable outcomes in clinical trials [208,210]. Considered as a more superior SSRI, escitalopram oxalate exhibits favorable tolerability. However, the understanding of the mechanism of how escitalopram oxalate suppresses GBM is still limited, which needs more research to investigate the potential use in GBM [167]. Researchers have been testing the idea of drug repurposing and evaluating drugs, and SSRIs are one of the most promising kinds.

### 4.6. Norepinephrine/Noradrenaline (NE/NA): Inhibits GBM Invasiveness and Migration, as Well as Modulates Tumorigenesis

In the past few decades, β-AR agonists have been indicated as one of the central modulators of tumor initiation and progression by triggering the cyclic adenosine monophosphate/protein kinase A (cAMP/PKA) and MAPK signaling. Although the precise mechanism is still unknown, it is becoming clearer as research advances [63,64]. He, J.J., et al. showed that activating β-ARs could upregulate MMP-2 and MMP-9 levels and trigger the ERK1/2 pathway, promoting U251 cell growth [128].

The level of Cx43 and gap junction intercellular communication form a transit system, which is able to amplify drug effects from a single cell to the whole tumor. Studies have demonstrated that β2-AR agonists upregulate the Cx43 protein levels in the GBM cell lines and promote cytotoxic effects [211,212].

Research conducted in astrocyte cells derived from human glioblastoma indicates that clenbuterol hydrochloride, a β2-AR agonist, upregulates the Cx43 expression level. In contrast, ICI 118551, a β2-AR antagonist reverses this activity. Moreover, both drugs have similar effects on co-culture systems. This new finding sheds light on a new approach to sensitizer design, which is to upregulate Cx43 expression levels by activating β2-AR [153].

As we discuss, activating β-ARs may result in anti-tumor effect (upregulate Cx43 protein levels) or pro-tumor effect (upregulate MMP-2 and MMP-9 levels). More research in this field would be of great help.

### 4.7. Purinergic: Involved in Tumorigenesis, GBM Invasion, Migration

Commonly, the expression of adenosine (P1) and nucleotide (P2) receptors is found not only on neurons, but also on glial cells. As a major regulator in presynaptic activities, P1 receptors serve as critical elements in the crosstalk among tumor cells, astrocytes and microglia, influencing tumor-related astrocyte phenotypic differentiation [66]. On the one hand, P2X ionotropic receptors participate in synaptic remodeling and rapid synaptic transmission. P2Y receptors, on the other hand, are implicated in presynaptic modulation, which includes inducing long-term (nutritional) signaling in the cell life circle. P1 and P2 receptors are both related to communication between neurons and glia cells, which modulate neuronal activities involved in inflammatory reaction, myelination, tumorigenesis and neurodegeneration [213].

Several articles published recently indicate that the elimination of extracellular ATP and the increase in adenosine directly not only affect tumor progression, invasion and angiogenesis, but also induce the migration, adhesion and homing between cancer cells, activating the immune system [214]. Activating P2X7R triggers various effects including cell growth, death and immune cell activation, depending on different kinds of microenvironment and expression levels [215,216]. P2X7R is associated with insensitivity to ATP cytotoxicity, which has been proven in research conducted in vitro by Morrone FB [217]. Research also revealed that in glioma treatment, a higher P2X7R level represents a better outcome in radiotherapy and a longer survival time. Data analysis on P2X7R articles suggests its promising future applied in predicting radiotherapy outcomes and survival probability [218]. The application of BBG (a P2X7R inhibitor) remarkably suppresses the growth of C6 tumors. However, in the U251 glioblastoma cell line, the result is just the opposite. [155]. This comparison suggests that more research is needed concerning the drug design of P2X7R.

Rhizoma Paridis (RP; the rhizome of Paris) has been proven to suppress tumor growth. Research published previously demonstrated that saponins extracted from RP has wide anti-cancer effects, such as mediating apoptosis and interfering the cell cycle, activating the immune response and inhibiting migration with angiogenesis. Paris saponin H (PSH), a steroid saponin molecule of RP, is similar to Paris Saponin II, Polyphyllin D and Polyphyllin VII. Paris Saponin II inhibits angiogenesis by downregulating VEGF and disturbing signaling involving VEGFR2 [219]. Polyphyllin D and Polyphyllin VII are associated with cell cycle arrest and apoptosis procedures [220,221]. Research conducted on mice has proven that total steroidal saponins can activate the receptors of adenosine, and some saponins (such as 8-cyclopentyl1, 3-(N)-xanthine, 3-dipropyl-2 and N6-cyclohexyladenosine) are capable of blocking the A1/A3 adenosine receptor [156].

We notice the complexity in transforming basic trials of purinergic activities into practical applications. However, it is worth exploring to achieve a deeper understanding.

### 4.8. Neuropeptide

#### 4.8.1. Neuropeptide Y (NPY): Stimulates Tumor Growth and Progression

NPY, secreted by hippocampal interneurons, is a type of abundant neurotransmitter in CNS [222,223]. It has been confirmed that NPY has multiple effects related to cancer biology, such as stimulating tumor-associated angiogenesis, cell proliferation, migration and chemoresistance, as well as regulating the differentiation of stem cells [68]. NPY may have special effects on GBM. An elegant study shows that NPY could not be seen in many brain tumors but could be found in nerve fibers of most kinds of glioma. Furthermore, the expression of Y2 receptors on GBM cells is upregulated and distributed extensively [69]. More research in vitro and in vivo is needed to confirm the regulation of NPY on GBM. In addition, leptin is well-known to inhibit NPY neurons in the arcuate nucleus of the hypothalamus [224]. However, since the exact molecular mechanism remains unclear, there is still a long way to go before leptin could be put forth into clinical development.

Recent studies have tried to utilize NPY in cancer imaging and therapy, focusing on breast cancer first [222]. hY1R-preferring agonists, especially [F^7^, P^34^]-NPY, have shown great potency in imaging tumor sites in patients with breast cancer, thanks to the high frequency and density of Y1 receptors on breast tumor cells, as well as the discrepancy of the receptor subtypes between breast carcinomas (Y1 receptors) and non-neoplastic breast (Y2 receptors) [225,226]. Another novel advancement is peptide-drug conjugates (PDC), which serve as a new carrier promising to deliver antitumor drugs to tumors selectively without damaging healthy tissues. Consistently, this kind of conjugate is dependent on the specific expression of cell surface receptors as well [227]. In GBM, the high frequency and density of Y2 receptors may also create a condition in favor of selective GBM targeting [69,227]. Nevertheless, the high expression of Y2 receptors is compared with other human tumors, and there is insufficient evidence to support the difference in NPY receptor subtypes between GBM and normal brain tissues. Further investigation is required.

#### 4.8.2. Substance P (SP): Induces GBM Proliferation, Invasiveness and Migration

SP, which is a neuropeptide released by neuronal and non-neuronal cells, belongs to the family of tachykinins [228]. The overexpression of NK-1Rs can be found both on the tumor cell membranes and in the neovasculature of the GBM microenvironment, and results in a worse survival time and prognosis for GBM patients [71,72]. Activating NK-1Rs leads to the phosphorylation of AKT and ERK1/2, the activation of c-myc and activator protein-1 (AP-1) and the promotion of DNA synthesis [100,229]. As a result, this type of receptor emerges as having critical participation in the proliferation, invasion and migration of GBM, via inducing anti-apoptotic effects, favoring angiogenesis and triggering inflammation correlated with tumor progression [73,74,75]. NK-1Rs have also been confirmed to have a direct impact on the migration of GBM via upregulating MMP-2 and MT1-MMP [76]. In addition, the pathological transformation of the original signal transduction pathway in astrocytes may correlate with the upregulation of NK-1R [230].

Of note, the effects of NK-1R antagonists on cell growth and apoptosis have been identified and summarized recently [73]. In addition, an innovative co-treatment with NK1R antagonists and CaMKII inhibitors has shown synergistic inhibition on the growth of GSCs and their derivatives both in vitro and in vivo [231]. Further experiments on animal models in the future will make efforts to translate this combination therapy to the clinic.

Therapeutics that deliver radiation at toxic levels to the site of the disease by a radiolabeled molecule are known as targeted radionuclide therapy. Due to the high and specific expression of NK-1Rs on GBM cells, injecting SP analogues labeled with β^−^-/α- emitting isotopes locally represents a novel therapy for GBM, especially in tumor recurrence [71,232]. It is proved that α particles (^225^Ac, ^213^Bi, ^211^At) are more effective and specific than β− particles (^131^I, ^90^Y, ^177^Lu) [71]. The experiment in GBM cell lines and the GSC line has demonstrated the inhibition of ^225^Ac-1,4,7,10-Tetraazacyclododecane-1-Glutaric Acid-4,7,10-Triacetic Acid (DOTAGA/DOTA) -SP on the viability, proliferation and apoptosis of tumor cells [233]. In addition, several clinical studies have indicated that targeted radiopeptide therapy is safe and well-tolerated, concerning 90Y-DOTA-SP [71,234], ^213^Bi-DOTA-SP [71,235,236,237], ^225^Ac-DOTA-SP [232] and ^211^At-DOTA-SP [238]. Taken together, it is expected that SP analogues and NK-1R antagonists could be feasible for controlling GBM in the future.

### 4.9. NO: Involved in GBM Proliferation and Migration and Cell Apoptosis

NO is produced by three subtypes of nitric oxide synthase (NOS), including neuronal NOS (nNOS), inducible NOS (iNOS) and endothelial NOS (eNOS) [239]. An immunohistochemistry check for NOS expression reveals that, in contrast to healthy human brains, NOSs are overexpressed in gliomas and vary in different types and levels of this kind of tumor. In total, systematic analysis shows the upregulation of NOSs in glioma cells and the peritumor microenvironment, suggesting that NO production is complex and contains various sources [81].

Scholars have held different opinions on the role of NO in cancer biology, namely whether it functions as a tumor promoter or inhibitor [84]. This discussion could be attributed to different research backgrounds, such as the tumorigenesis stage, NO expression level, the NOS subtype and its location and various kinds of neoplastic cells and their differentiation status [81]. Among all these variables, the concentration may matter most. Some suggest that a moderate concentration (100–350 μmol/L) reduces cell proliferation and genomic instability, while some of the literature shows that a high concentration (500–1000 μmol/L) causes cell reproduction decline, apoptosis upregulation, irreversible DNA impairment and Ataxia telangiectasia mutated/Ataxia telangiectasia and Rad3-related protein (ATM/ATR)-dependent p53 phosphorylation induced by downstream signaling pathways [78,79,80,81,82,83].

EGFRvIII/STAT3 pathway is critical signaling downstream in glioblastoma and gets involved in GBM cell proliferation, migration and progression. In addition, STAT3 activates iNOS via binding to the promoter of iNOS genes. Therefore, inhibiting iNOS pharmacologically and genetically can suppress invasiveness and reproduction of GBM cells. 1400W and S-MIU are selective iNOS inhibitors [72,73,74,75] that reduce cell proliferation of astrocytes and U87 cell lines with EGFRvIII expression [84].

As mentioned before, NF-KB is involved in malignance formation through regulating downstream pathways. There is research conducted in C6 glioma cells indicating that NF-kB modulation is significant to Nos2 regulation. NF-KB activation is associated with a nondegradable form of IjBa, which interferes with NF-KB signaling and LPS, as well as the expression of cytokine-stimulated iNOS [240].

Reviewing back, the key point in understanding the role of NO has been summarized down to the NO concentration, the redox environment and the exposure time [239]. Considerably, more work will need to be done to identify the effect of NO.

## 5. Conclusions and Future Directions

As shown above, the influence of neurotransmitters on GBM is complicated: different transmitters activate different pathways, and even the same transmitter can cause the opposite effect due to the diversity of receptor subtypes and environments. For example, as mentioned before, the role of NO in tumor biology can be pro- or anti-tumor activities, which may be caused by multiple factors, including the level of NO/NOS in the peritumor microenvironment, the NOS type, the tumor stage and the degree of the differentiation of the tumor, as well as the tumor subtypes that express NOS preferentially [81]. Therefore, while testing new drugs, it is of great significance to design tailored clinical trials according to the characteristics of each neurotransmitter [5].

In addition to assisting in synaptic transmission, neurotransmitters act on a wide variety of cells in the GBM microenvironment, including neurons, tumor cells, glial cells and vascular endothelial cells, thus, mediating a broad regulatory effect on the survival, progression, recurrence and chemo-/radio-resistance of GBM. Hence, it can be expected that targeting neurotransmitters could be a promising novel pharmacological approach for GBM treatment. For one thing, taking into account drugs associated with neurotransmitters provides us with an abundant drug box. For another, the capability of GBM cells to alter the expression levels of neurotransmitter receptors allows them to be distinguished from non-tumor cells in the microenvironment and specifically targeted by drugs, thus, reducing side effects [227].

Specifically, therapeutic options associated with neurotransmitters may include: (i) activating/silencing numerous neurotransmitters receptors or regulating the number of receptors via gene expression. Notably, in pursuit of high efficiency and low toxicity, receptors that are differently expressed in tumor cells are considered to be more suitable as molecular targets, such as α9nAChRs, NK-1Rs and Y2Rs; (ii) providing a novel molecular marker when evaluating therapeutic strategies and the prognosis of patients, such as α9nAChRs that are up-regulated in GBM and predict a negative prognosis [241]. In addition, Yuri Belotti et al. have identified a subset of ten neurotransmitter receptors genes (DRD1, HTR1E, HTR3B, GABRA1, GABRA4, GABRB2, GABRG2, GRIN1, GRM7 and ADRA1B) that correlates with the unfavorable prognosis of GBM [242]; (iii) regulating the metabolism of neurotransmitters (synthesis, storage, release and inactivation), such as Amb4269951, SSZ, Escitalopram oxalate, 1400W and S-MIU; (iv) utilizing neurotransmitters or their receptor as drug carriers, such as nAChR-mediated transcytosis and PDC; (v) applying with other types of drugs (such as TMZ) to amplify efficacy or reduce chemoresistance. In order to increase the accuracy of targeted therapies and lower side effects, more analysis of differential gene expression between GBM and normal cells is needed.

Although the drugs mentioned in our research are mainly applied in vitro, there are several drugs in our studies that have progressed to phase I/II clinical trials, which include memantine [159], talampanel [160,161,162], ONC201 [164,165], chlorpromazine [166] and escitalopram oxalate [167]. Furthermore, experiments in mouse models also show good results [41,127,149,150,157,158]. Hence, we hold an optimistic perception that, with further research, drugs targeting neurotransmitters may be applied in GBM treatment. In order to reduce the duration and cost of drug development, we suggest that future work could give priority to the repurposing of FDA-approved drugs [208,243]. In general, those compounds consist of small molecule inhibitors and chemically modified derivatives of drugs, with high bioactivation in vivo and concerning blood-brain barrier (BBB) penetrability [244], such as psychotropic, antiepileptic, antidepressive and antihypertensive drugs [159].

Furthermore, different molecular subtypes of GBM may be sensitive to different neurotransmitters since they contain different mutations. For example, the activation of M2 receptors is likely involved in the mediation of Notch-1/EGFR pathways in p53 wild-type GB7 cells, which could not be seen in p53-deficient GB8 cells. Instead, M2 receptors on p53-deficient GB8 cells may induce a cytotoxicity effect mediated by oxidative stress and apoptosis [13,134]. As such, the effect of targeting M2 receptors for treatment correlates with the genotype of patients and the molecular subtypes of GBM [13]. To achieve favorable efficacy, it is vital to detect the genotype of the patients and the molecular subtypes of GBM when considering drug options. Future studies should explore targeted drugs combined with genomics and proteomics analysis, in order to accomplish precision medicine and improve the prognosis of GBM patients [245].

## Figures and Tables

**Figure 1 cancers-14-03970-f001:**
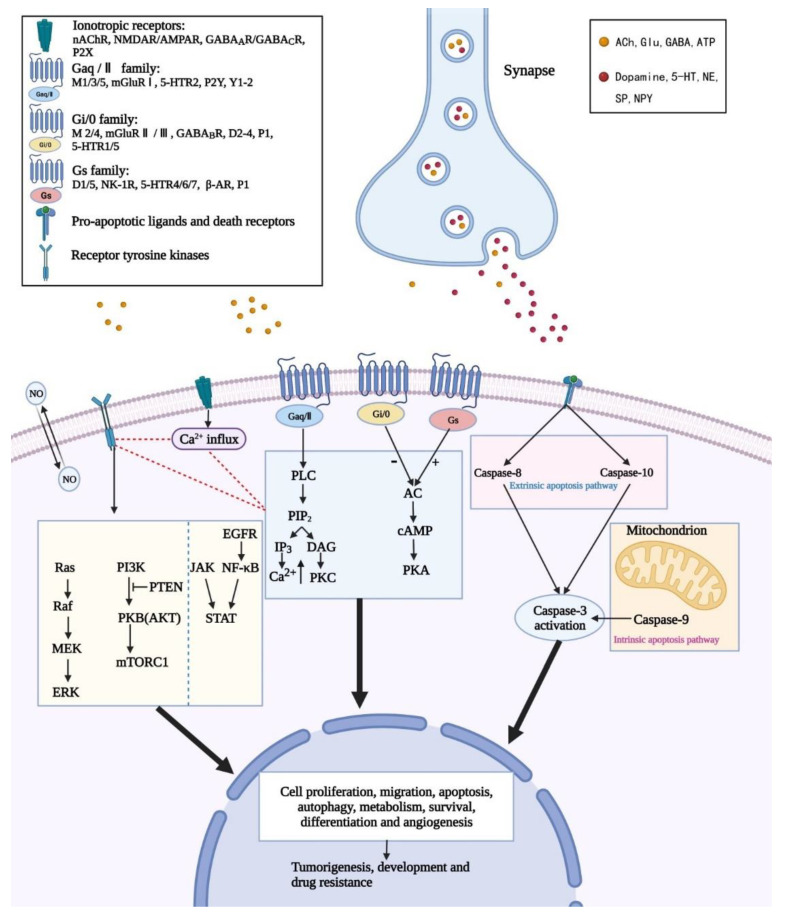
The modulation of neurotransmitters on common molecular pathways. Neurotransmitters mediate the transformation of extracellular ligand signaling into intracellular signaling through multiple kinds of receptors, thus, affecting cell proliferation, migration, metabolism, apoptosis, autophagy, survival, differentiation and angiogenesis. The main intracellular signaling includes RTK pathways, GPCR pathways, apoptosis and autophagy signaling pathways. The activation of RTKs and GPCRs are associated with Ca^2+^ flux, which has been considered as the initial impulse and critical modulator of GBM signaling. All neurotransmitters can be identified by GPCRs, while ACh, Glu, GABA and ATP secretion activate Ca^2+^ flux via ionotropic receptors. GPCR mainly consists of three families: (i) Gaq/11 family (including M1/3/5 mAChR, mGluR I, 5−HTR2, P2Y and Y1/2), (ii) Gi/o family (including M2/4 mAChR, mGluR II/III, GABABR, D2−4, P1 and 5−HTR1/5) and (iii) Gs family (including D1/5, NK−1R, 5−HTR4/6/7, β−AR and P1). The mTOR pathway plays a vital role in autophagy induction. NF−kB has been shown to get involved in tumorigenesis, tumor growth and tumor response to drugs as a transcription factor. Apoptosis is mainly related to two mechanisms: (i) the extrinsic pathways initiated by pro-apoptotic ligands and (ii) the intrinsic pathways mediated by mitochondrial permeabilization. Neurotransmitters become involved in the activation of pro−caspase−8/9/10/3.

**Figure 2 cancers-14-03970-f002:**
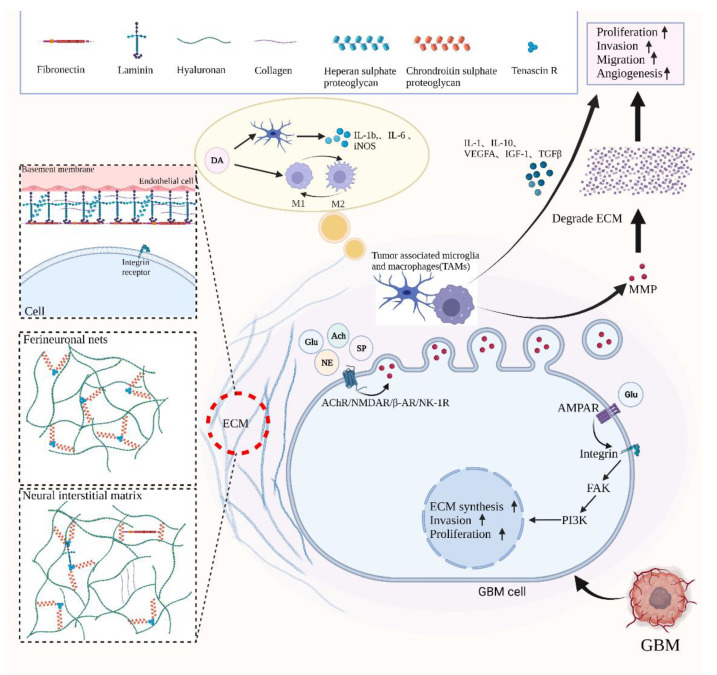
The role of neurotransmitters in regulating infiltrative capacity of GBM. Neurotransmitters, such as ACh, Glu, NE and SP, regulate invasion behavior of GBM by affecting the activation of matrix metalloproteinases (MMPs). MMPs are a group of proteolytic enzymes secreted by cancer cells and have the ability for dissolving various kinds of extracellular matrix (ECM). The ECM, containing various kinds of molecules, is mainly localized to three compartments: the basement membrane (basal lamina), the perineuronal nets and the neural interstitial matrix. MMPs influence the lesion’s microenvironment during remyelination by cleaving CSPGs, laminins and ECM receptors such as integrins. GBM cells prefer to migrate to a more rigid microenvironment, resulting in the impairment of tissue morphology and promoting tumor growth and development. The hardness of the microenvironment is in relation to the composition of the ECM: both the enhanced matrix crosslink in the ECM and the anchoring of integrin-focal adhesion complexes (FAK) between cells and the ECM have provided the desired properties to allow tumor invasion. Tumor-associated microglia and macrophages (TAMs) participate in immunomodulation and MMP secretion.

**Figure 3 cancers-14-03970-f003:**
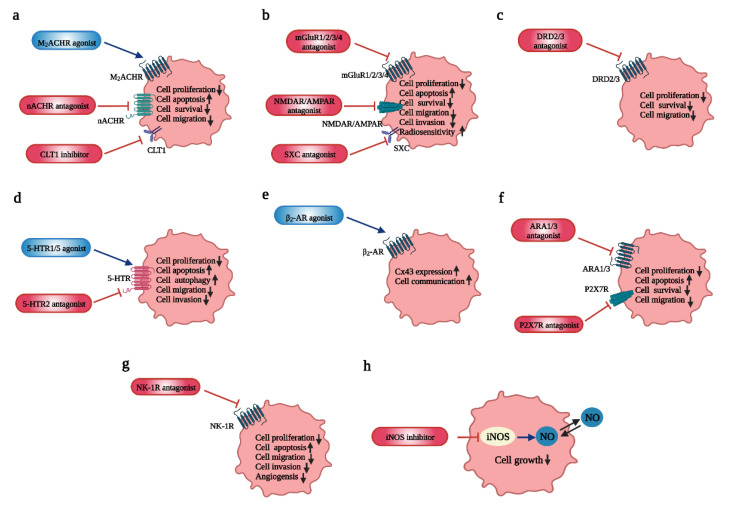
Potential targets of neurotransmitter-related drugs in GBM. Different neurotransmitters and their potential targets for GBM treatment. (**a**) ACh and M2AChR, nAChR, CTL1. (**b**) Glu and mGluR1/2/3/4, NMDAR/AMPAR, SXC. (**c**) DA and DRD2/3. (**d**) 5-HT and 5-HTR1/2/5. (**e**) NE and β2-AR. (**f**) ATP and ARA1/3, P2X7R. (**g**) SP and NK-1R. (**h**) NO and iNOS inhibitor.

**Table 1 cancers-14-03970-t001:** The function of neurotransmitters in CNS and GBM.

Neurotransmitters	Receptors	Physiological Functions	References	Pathological Functions	References
Acetylcholine	mAChR M1–M5 nAChR	- modulates the activity of NSCs in adults’ hippocampus, SVZ and SGZ	[20,21,22]	- regulates GBM proliferation, survival and invasion	[27,28,29,30,31,32,33,34,35,36]
Glutamate	NMDAR AMPAR mGluR I–III	- modulates the activity of neurons and the plasticity of synapses - induces cell proliferation, migration, survival and cell differentiation of NPCs - regulates embryonic neurogenesis and migration of neurons and astrocytes	[23,37,38,39]	- modulates GBM growth and progression - induces excitotoxic cell death of neurons - increases the radiosensitivity of GBM	[40,41,42,43,44,45,46,47,48,49,50]
GABA	GABA_A_R GABA_B_R GABA_C_R	- regulates the excitability of neurons and NPCs - increases proliferation and migration of both embryonic cells and neuroblasts in postnatal SVZ - inhibits the proliferation of astrocytes - modulates both the differentiation and the quiescence of NPCs - controls the migration and cell proliferation of NPCs	[25,51]	- inhibits GBM growth	[12,52,53,54]
Dopamine	DRD1-DRD5	- participates in reward, motor control and motivation	[55,56,57]	- modulates tumor progression - inhibits tumor migration by reducing stress-mediated angiogenesis	[58]
Serotonin	5-HTR 1–7	- contributes to the completeness of neuronal circuits and synaptic function	[59]	- triggers autophagy and apoptosis of GBM cells	[60]
Norepinephrine/ Noradrenaline	β-AR	- inhibits NPCs proliferation and neurogenesis	[61]	- inhibits GBM cells invasion and migration - modulates tumor initiation and progression	[62,63,64]
Purinergic	P1,P2	- involved in memory, study, diurnal circle, locomotor and feeding activity	[65]	- participates in tumor remodeling and cerebral vascular tone regulation - influences tumor-related astrocytes phenotypic differentiation	[66]
Neuropeptide Y	Y1, Y2, Y4, Y5	- regulates the activity of neurons - protects for neurons by inhibiting the excitotoxic concentration of glutamate - regulates the proliferation, differentiation and migration of NSCs in SVZ and dentate gyrus	[24,67]	- stimulates tumor growth and progression	[68,69]
Substance P	NK-1R	- promotes the proliferation and differentiation of NSCs - induces NSCs to differentiate into neurons more than into astrocytes	[70]	- induces GBM proliferation, invasiveness and migration	[71,72,73,74,75,76]
NO	-	- regulates neuron reproduction, differentiation and life span - involved in synaptic functional activity, memory storage and neural remodeling	[77]	- involved in GBM proliferation and migration and cell apoptosis	[78,79,80,81,82,83,84]

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
