# Peer review of "Neurotransmitters: Potential Targets in Glioblastoma"

_cancers, 2022, doi:10.3390/cancers14163970_

Round 1

Reviewer 1 Report

In the manuscript titled “Neurotransmitters: Potential Targets in Glioblastoma” prepared by Huang et al., the author discussed the role of neurotransmitters, a common element in the brain tumor microenvironment, in the disease process of GBM. I have several concerns as follows:

1. The topic of the present review is interesting, although there are many issues with the proper use of the language and formatting. I would recommend the author seek help from an editing service from a native speaker.

2. Since neurotransmitters are very important in normal brain signal transduction, what is the specific way to target the neurotransmitters in GBM other than in the normal brain? The side effect will be an obstacle for the normal brain if the target is not tumor-specific. And several drugs of neurotransmitters in a clinical trial are drugs that targeted depression in the past, will they cause side effects on mental health?

3. Since targeting glioma by neurotransmitters or receptors expressing specific in GBM is more promising, targeting tumor favorable neurotransmitters or receptors such as targeting α9nAChRs seems to have better chances for glioma, and it’s better to have more discussion on these kinds of treatment.

4. In part 2, the title is “neurotransmitters function as necessary components”, but the content is about the introduction of the neural microenvironment, GSC, and GBM microenvironment, with few details about the importance of neurotransmitters, especially in GBM.

5. The manuscript needs some proofreading. Some of the typing, abbreviations, and references need to be corrected. Such as the title of “5.5” to “5” in line 718. Abbreviations such as DRD2 and NMDAR are not presented fully. “Error! Reference 358 sources not found” can be seen in lines 358-359.

6. The author may also consider discussing the molecular subtypes within GBM. For example, the patterns in the genetic landscape define several sub-cluster in GBM (PMID: 24120142). EGFR amplified (PMID: 11504770), PI3K mutated (PMID: 15924253), or IDH mutated (PMID: 32825279) GBM may exhibit very different dependency on neurotransmitter stimuli.

Reviewer 2 Report

In this manuscript, the authors summarized in detail the relationship between neurotransmitters and the initiation, progression, and invasion of GBM, as well as the research progress of drugs targeting neurotransmitters. The cited references are detailed, the manuscript was refined reasonably, and the language was qualified. I think it is a worthwhile review, which has a strong attraction to readers. Here are some suggestions:

1.      Summary the types, subtypes, physiological and pathological functions of neurotransmitters in a list for better understanding of readers.

2.      Supplement the details about immune cells infiltration in Figure 2.

3.      It is suggested to display the targets of neurotransmitter-related inhibitors/drugs in a figure.

4.      The authors dedicated a significant portion on molecular functions and the research status of inhibitors/drugs of neurotransmitters in section 4. I suggest that the authors could put the functional part into the section 3 for the balance of each section of manuscript.

Round 2

Reviewer 1 Report

The author made improvements to the manuscript.

Reviewer 2 Report

Good enough for acceptance.